# Papain-Like Proteases as Coronaviral Drug Targets: Current Inhibitors, Opportunities, and Limitations

**DOI:** 10.3390/ph13100277

**Published:** 2020-09-28

**Authors:** Anastasiia I. Petushkova, Andrey A. Zamyatnin

**Affiliations:** 1Institute of Molecular Medicine, Sechenov First Moscow State Medical University, 119991 Moscow, Russia; asyapeti@gmail.com; 2Belozersky Institute of Physico-Chemical Biology, Lomonosov Moscow State University, 119992 Moscow, Russia; 3Department of Biotechnology, Sirius University of Science and Technology, 1 Olympic Ave, 354340 Sochi, Russia

**Keywords:** coronavirus, papain-like proteases, inhibitor, outbreak

## Abstract

Papain-like proteases (PLpro) of coronaviruses (CoVs) support viral reproduction and suppress the immune response of the host, which makes CoV PLpro perspective pharmaceutical targets. Their inhibition could both prevent viral replication and boost the immune system of the host, leading to the speedy recovery of the patient. Coronavirus disease 2019 (COVID-19) caused by severe acute respiratory syndrome coronavirus-2 (SARS-CoV-2) is the third CoV outbreak in the last 20 years. Frequent mutations of the viral genome likely lead to the emergence of more CoVs. Inhibitors for CoV PLpro can be broad-spectrum and can diminish present and prevent future CoV outbreaks as PLpro from different CoVs have conservative structures. Several inhibitors have been developed to withstand SARS-CoV and Middle East respiratory syndrome CoV (MERS-CoV). This review summarizes the structural features of CoV PLpro, the inhibitors that have been identified over the last 20 years, and the compounds that have the potential to become novel effective therapeutics against CoVs in the near future.

## 1. Introduction

Coronaviruses (CoVs) have been responsible for three viral pandemics in the last 20 years. For SARS (severe acute respiratory syndrome), 8096 cases (mortality = 9.6%) were registered in 2002–2003 according to the World Health Organization (WHO). MERS (Middle East respiratory syndrome) was identified in 27 countries and resulted in 2494 confirmed cases (mortality = 34.4%) since 2012. As of August 2020, 18,575,326 cases (mortality = 3.8%) of CoV 2019 (COVID-19) caused by SARS-CoV-2 were registered in 216 regions (WHO; URL: https://www.who.int/). The percent death rates for the first two CoV outbreaks were higher; however, previous CoVs caused less severe impacts on the world as they resulted in fewer cases of disease [1]. Unlike MERS-CoV, the viruses SARS-CoV-2 and SARS-CoV are easily transmitted between humans [2], and, in contrast to patients with SARS-CoV, individuals with COVID-19 are contagious before they show the first symptoms of viral infection [3]. These differences are explained by the distinct cellular receptors to bind [4] and by various structural features of S-proteins [5]. These factors have contributed to the rapid spread of COVID-19. There is no approved treatment for CoV-related diseases although a variety of compounds are currently being investigated to suppress CoV disease in clinical trials (COVID-19 Vaccine & Therapeutics Tracker; URL: https://biorender.com/covid-vaccine-tracker).

One approach to suppress a disease is to inhibit the proteases promoting it. Papain-like cysteine proteases (PLCPs) comprise the peptidase family C1 of clan CA. They have structures similar to papain from *Carica papaya* and possess cysteine, histidine, and asparagine in the catalytic center [6]. They have been found in most living organisms and conduct multiple functions, such as matrix turnover and antigen presentation in animals [7] and senescence and regulated cell death in plants [8,9]. This made them perspective targets for inhibition and induced intensive study of PLCPs over the last years. Multiple inhibitors have been developed for human cysteine cathepsins, which play a crucial role in tumor progression [10,11,12], and for pathogen PLCPs, such as cruzipain from *Trypanosoma cruzi* and falcipain from *Plasmodium falciparum* [13,14].

PLCPs have also been found in hepatitis E, enterovirus G, and CoVs, where they mainly digest viral polyproteins and interfere with the host immune response [15,16]. Despite the heightened interest in viral papain-like proteases (PLpro) and multiple designed inhibitors for CoVs [17], there is still no approved drug. Apparently, as for other antivirals, the main two challenges in the development of CoV inhibitors are drug delivery into the human cells and the frequent mutations of viral genomes, which impede the development of a broad-spectrum inhibitor [18]. In this review, we attempted to highlight both the previous successes and the gaps in the development of the inhibitors against CoV PLpro and to ascertain future challenges on the way to a safe and effective therapeutic.

## 2. Coronaviruses

SARS-CoV, MERS-CoV, SARS-CoV-2, and other human CoVs (HCoVs), such as HCoV-229E, HCoV-NL63, HCoV-OC43, and HCoV-HKU1, relate to the family *Coronaviridae* (suborder *Cornidovirineae*, order *Nidovirales*, realm *Riboviria*), which comprises 39 species in five genera [19] that were first identified 60 years ago [20]. They spread mainly via respiratory and fecal–oral routes among mammals and birds, causing mainly respiratory, enteric, hepatic, and neurological diseases. CoVs are readily transmitted across species as their ribonucleic acid (RNA) tends to undergo frequent recombinations and mutations [21].

CoVs comprise a positive-sense single-stranded RNA which is the largest RNA genome (26–32 kilobases) among all the identified viruses that encode 15 or 16 proteins of the replication-transcription complex (RTC), four or five structural proteins, and one to eight group-specific or accessory proteins [22]. CoV genome is enveloped in the membrane, which contains the transmembrane (M) glycoprotein, the “crown”-like spike (S) glycoprotein, and the envelope (E) protein [23]. CoV infection is initiated by the binding of S proteins on the surface of the virion with the cellular receptors aminopeptidase N (APN) for HCoV-229E; angiotensin-converting enzyme 2 (ACE2) for SARS-CoV-2, SARS-CoV, and HCoV-NL63; and dipeptidyl peptidase 4 (DPP4) for MERS-CoV [4,24]. This causes the deposition of the nucleocapsid into the cytoplasm where the viral genome is translated. The positive-sense genome serves as the first messenger RNA (mRNA) of infection, which is translated into the enormous replicase polyprotein. Then, the RTC synthesizes the progeny genomes and a set of subgenomic mRNAs. The latter are translated into structural proteins and accessory proteins [25]. Invasion, replication, and host immune response suppression by CoV require numerous viral and host enzymes.

Moreover, 16 nonstructural proteins (nsps 1–16) are encoded in the CoV genome [25], which include several enzymes, such as papain-like protease (PLpro), 3-chymotrypsin-like protease (3CLpro), helicase, RNA-dependent RNA polymerase (RdRp), and primase [26] (Table 1). The viral RNA is translated into polyproteins, which are then cleaved into effector proteins by the viral proteases 3CLpro and PLpro [27]. PLpro also possess deubiquitinase activity and it was shown that they may deubiquinate interferon (IFN) regulatory factor 3 (IRF3) and nuclear factor kappa-light-chain-enhancer of activated B cells (NF-κB) of the host cell, resulting in immune suppression [28]. The proteases release the enzymes that are important for the viral RNA replication: helicase, RdRp, and NTPase. These are integrated into a membrane-associated viral enzymatic RTC, which plays the primary role in the replicative cycle of RNA viruses [29]. The RTC produces the negative-strand RNA and subgenomic mRNAs [30]. The replication is of relatively low accuracy, which is believed to be the major factor in the evolution, adaptation, and epidemiology of RNA viruses [31].

CoVs also recruit host cell enzymes. As the viral genome does not encode a translational apparatus, CoVs need the host one to produce their proteins [32]. Apart from this, endosomal human proteases promote the invasion of the virus. Successful viral infection requires the S protein to be activated by proteolysis before or after the binding to the cellular receptor. This induces the fusion of the viral and cellular membranes. The host proteases involved in this process are the cathepsins B and L, type II transmembrane serine proteases (TMPRSS), the human airway trypsin-like protease (HAT), trypsin, and cathepsins B, L, and S, etc. [33]. In summary, the enzymes regulate the main stages of the CoV life cycle, including invasion, translation, polyprotein digestion, replication, and transcription. Therefore, the enzymes of CoVs are prime targets for the development of inhibitors (Table 1).

## 3. CoV Enzyme Inhibitors and Their Feasibility for Therapeutics

Inhibitors have been developed for every enzyme encoded by the CoV genome, except for one of the viral-encoded endoribonucleases (nonstructural protein 1, nsp1) and primase. Multiple inhibitors prevent the first stage of CoV infection—the invasion of the virus. This group of inhibitors usually targets either cathepsin L or TMPRSS2. Tetrahydroquinoline oxocarbazate (CID 23631927) is a subnanomolar, slow-binding, reversible inhibitor of human cathepsin L, which blocks SARS-CoV and Ebola virus entry into human cells [34]. SSAA09E1 inhibits cathepsin L and, as a result, blocks SARS-CoV entry [35]. High-throughput screening (HTS) based on peptides, derived from the structures of glycoproteins of SARS-CoV, Ebola, Hendra, and Nipah viruses, was used to discover a small molecule, 5705213, and its derivative 7402683, which inhibit the cathepsin L cleavage of SARS-CoV spike glycoprotein in an in vitro assay [36]. HTS of Food and Drug Administration (FDA)-approved drugs was used to identify that teicoplanin, a glycopeptide antibiotic, and its derivatives, such as dalbavancin, oritavancin, and telavancin, which potently prevent the entry of Ebolavirus, MERS-CoV, and SARS-CoV into the cytoplasm via the inhibition of cathepsin L. Notably, teicoplanin has routinely been used in clinical applications with low toxicity [37]. A commercial serine protease inhibitor (camostat) partly blocks the infection by SARS-CoV in HeLa cells. The full blockage was achieved by simultaneous treatment of the cells with camostat and EST, a cathepsin inhibitor [38]. Together with the inhibitors for cathepsin L, such as K11777 and vinyl sulfones, camostat blocks the invasion of SARS-CoV, MERS-CoV, and HCoV-NL63 [39]. K11777 is a safe and effective cysteine protease inhibitor, which was identified as a therapeutic agent for parasitic diseases, such as Chagas disease. Subnanomolar amounts of K11777 prevent SARS-CoV and Ebola virus entry [39]. At the moment, the impact of Camostat Mesylate on COVID-19 infection is being investigated in phase II clinical trials.

The invasion of CoV causes the entrance of the viral genome into the host cell. The viral RNA encodes the enzymes of the RTC. Their inhibition can prevent viral replication and, as a result, the assembly of the novel virions. Various nucleotide derivatives, including remdesivir, sofosbuvir, ribavirin, and favipiravir/favilavir, have been suggested for use as inhibitors of the viral RdRp [40,41,42]. Remdesivir is a prospective drug for CoV treatment but others have shown either controversial results on efficiency or adverse effects [43]. A number of benzotriazole, phenothiazine, triphenylmethane, acridone, small peptide, etc. derivatives inhibit the helicase of the viruses from the Flaviviridae, Coronaviridae, and Picornaviridae families [44]. Fluorescence resonance energy transfer (FRET)-based dsDNA unwinding assay and a colorimetry-based adenosine triphosphate (ATP) hydrolysis assay revealed that naturally occurring flavonoids, myricetin and scutellarein, inhibit the SARS-CoV helicase protein in vitro by affecting the ATPase, but not the unwinding activity. These compounds did not show cytotoxicity against normal breast epithelial MCF10A cells [45]. A FRET-based helicase assay also identified the compound SSYA10-001, which specifically blocks the unwinding activities of SARS-CoV, MERS-CoV, and murine CoV (MHV) helicase, with low cytotoxicity [46]. The von Hippel Lindau protein (VHL) was proven to inhibit SARS-CoV replication by regulating viral 2′*O*-MTase ubiquitination and promoting its degradation [47]. Novel chemical inhibitors against SARS-CoV helicase, 7-ethyl-8-mercapto-3-methyl-3,7-dihydro-1H-purine-2,6-dione, and (*E*)-3-(furan-2-yl)-*N*-(4-sulfamoylphenyl) acrylamide suppress the ATP hydrolysis and DNA unwinding activities of helicase. Moreover, the latter compound did not show significant cytotoxicity [48]. Triazole derivatives inhibit both adenosine triphosphate (ATP)ase and unwinding activities of MERS-CoV helicase [49]. Sinefungin was proposed to inhibit methyltransferases of CoVs, whereas endonuclease (nsp15) is suppressed by RNase A inhibitors [50]. The enzymes involved in replication play a crucial role in the life cycle of CoVs, which makes their inhibition potentially effective. However, since their structures are similar to those of human enzymes, they may produce off-target effects [26,51].

Proteases are also essential for the viral life cycle and substrates of proteases are more diverse compared to the substrates of the enzymes involved in the replication. Thus, proteases represent more distinct targets for treatment, which decreases the probability of side effects. Moreover, peptide-like inhibitors are smaller and have less complicated structure compared to nucleotide-like inhibitors; thus, they are easier to synthesize and to deliver into a cell.

## 4. PLpro—A Prospective Pharmacological Target

The CoV genome encodes two proteases: PLpro and 3CLpro, which are responsible for cleaving nsp1 to nsp3 and nsp4 to nsp16, respectively, to generate the membrane-bound RTC [52]. These are both cysteine proteases, which are prospective targets for CoV treatment. However, unlike 3CLpro, PLpro has one more important function: it suppresses the host immune response.

Up to two PLpro are parts of the nsp3, which is the largest protein encoded by the CoV genome (200 kD) [53]. PL1pro was found in the alpha-CoVs and in clade A of beta-CoVs [54]; however, it is not complete or absent in other groups of CoVs. PLpro can reverse modifications by ubiquitin and interferon-stimulated gene 15 product (ISG15) [55]. Ubiquitination and ISGylation of viral and host proteins play essential roles in several immune pathways [56] that include the production and release of interferons (IFNs) within the IFN-I response and nuclear factor (NF)-κB within the inflammatory response, which is viewed as the first line of defense against viral infection [57]. CoV PLpro also affects the host innate immune response by blocking the phosphorylation and nuclear translocation of interferon regulatory factor 3 (IRF3) [58]. Thus, targeting CoV PLpro could impede both viral replication and immune suppression (Figure 1) inducing the immune response and patient recovery [59].

Multiple roles of PLpro in viral infection are mediated by its intricate structure. The folds of PLpro from different CoVs are quite similar (Figure 2). They resemble a right-hand fold which comprises three subdomains: the thumb and palm, where the catalytic triad is situated, and the fingers, which include the zinc-finger motif [53]. X-ray structures of PLpro from SARS-CoV, MERS-CoV, and transmissible gastroenteritis virus (TGEV) showed that a substrate-binding site is located between the thumb and the palm subdomains. The protease demonstrates a preference for cleaving substrates after the LXGG↓ sequence. The S1, S2, and S4 binding sites are conserved among PLpro, while S3 and S5 are slightly different [53]. In addition to the binding of the ubiquitin C-terminus to the substrate channel, there is an interaction between a hydrophobic region of SARS-CoV and MERS-CoV PLpro in the fingers subdomain and a hydrophobic patch of ubiquitin [60,61]. Since the structure of PLpro has been elucidated, this allows structure-based drug design. Moreover, since the folds of PLpro from different CoVs are similar, this protease is a prospective target for the development of a broad-spectrum inhibitor of CoV.

## 5. A Hindsight View of Developing CoV PLpro Inhibitors

Inhibitors of CoV PLpro identified to date include naphthalene and thiopurine derivatives, zinc conjugate inhibitors, and several natural products (Table 2) [28]. Zinc ions and their conjugates were the first identified inhibitors of SARS-CoV PLpro. However, the mechanism of inhibition has not been elucidated [62]. Zinc has also been shown to inhibit viral RdRp and human receptor ACE2 and to modulate the antipathogen immune and inflammatory response. This makes zinc a potential prophylactic and supplementary treatment [63]. However, its inhibitory activity against CoVs has not been investigated in clinical trials. SARS-CoV PLpro can be also inhibited by a series of natural products, such as tanshinones from *Salvia miltiorrhiza* [64], diarylheptanoids from *Alnus japonica* [65], geranylated flavonoids from *Paulownia tomentosa* [66], chalcones and coumarins from *Angelica keiskei* [67], and polyphenols from *Broussonetia papyrifera* [68]. These are reversible inhibitors that manifest different modes of inhibition: competitive, uncompetitive, mixed-type, and noncompetitive. Tanshinones inhibit the proteolytic and deubiquitinase activities of SARS-CoV PLpro with half-maximal inhibitory concentrations (IC_50_) of 0.8 and 0.7 µM, respectively [64]. Moreover, 8-(Trifluoromethyl)-9*H*-purin-6-amine is a reversible noncovalent inhibitor that is active against both SARS-CoV and MERS-CoV PLpro. It was identified as an allosteric inhibitor of SARS-CoV and a competitive inhibitor of MERS-CoV [69]. The cysteine protease inhibitor N-Ethylmaleimide (NEM) that covalently modifies the active-site Cys inhibited SARS-CoV, but was a poor inhibitor of MERS-CoV PLpro [70,71].

Additionally, 6-mercaptopurine (6MP) and 6-thioguanine (6TG) have been widely used in cancer treatment. They are competitive slow-binding inhibitors that form hydrogen bonds with the catalytic triads of SARS-CoV and MERS-CoV PLpro [70,71]. Disulfiram, a drug that has been used in alcohol aversion therapy since 1951, is an irreversible covalent inhibitor for SARS-CoV PLpro (competitive (or mixed)) and MERS-CoV (noncompetitive) [72]. The most potent inhibitors (according to IC_50_) of CoV PLpro are the naphthalenes identified by high-throughput screening (HTS) in 2008 [28]. The structure-activity relationship (SAR) analysis and the structural information obtained from the X-ray crystal structure of the most potent naphthalene inhibitor in complex with SARS-CoV PLpro allowed to design the noncovalent competitive inhibitors which suppress PLpro, with an IC_50_ of 0.15 µM [28].

To the best of our knowledge, naphthalene inhibitors are the only compounds that have been extensively studied and subject to lead optimization. However, attempts to inhibit MERS-CoV PLpro with naphthalene derivatives have failed [69]. About half of the identified inhibitors of CoV PLpro have not been assessed for activity against other CoVs, apart from SARS-CoV, even after the MERS-CoV and SARS-CoV-2 outbreaks. This has prevented the development of a broad-spectrum inhibitor. Moreover, several studies only assessed the inhibition of the protease activity of CoV PLpro, although it is also important to prevent deubiquitination and deISGylation of the proteins in the innate immune response by CoV PLpro (Table 2). This has interfered with the development of a therapeutic agent which could strike the virus at two key points in its life cycle. It is necessary to fill these gaps in the previous studies and to identify inhibitors that are active against various CoVs and suppress all enzymatic activities of the PLpro. The next step would be SAR analysis of the inhibitors and chemoinformatics to increase their effectiveness. This could be based on the experience of developing naphthalene inhibitors [28]. Another problem of the inhibitors of CoV PLpro is the lack of assays in cell cultures and cytotoxicity assessments. Since the structure of CoV PLpro is similar to those of human deubiquitinating enzymes, the inhibitors may cause adverse effects [53]. Additionally, 6MP, 6TG, and disulfiram appear safe as they are approved drugs. However, 6MP and 6TG are used in chemotherapy, and after the intracellular activation catalyzed by multiple enzymes, are cytotoxic via integration into DNA [73], an effect that can be intensified by the ability of 6TG to act as a UVA photosensitizer [74], whereas disulfiram irreversibly inhibits the activity of acetaldehyde dehydrogenase, leading to the accumulation of harmful metabolites [75]. Whether these activities preclude the use of these compounds as antiviral for CoV remains to be determined.

Why have most of these putative PLpro not been more thoroughly investigated? One reason may be that the previous CoV epidemics were local and affected relatively a few people, such that the interest in the search for therapeutics was limited to the period of the outbreaks and waned in-between (National Library of Medicine; URL: https://pubmed.ncbi.nlm.nih.gov/). For COVID-19, the high contagiousness and its rapid spread have led to a dramatic increase in the number of preclinical studies and clinical trials, which may eventually lead to the development of safe and effective drug treatment.

## 6. Current Development of Inhibitors of CoV PLpro

One logical approach to find CoV PLpro inhibitors for COVID-19 is to evaluate previously identified inhibitors of viral enzymes. The most rapid type of assay for screening thousands of compounds is high-throughput screening and virtual screening using molecular docking of such compounds into the binding sites of the enzymes [76,77]. The structure of the complex obtained either by docking or X-ray crystallography can shed light on the modes of binding, for further structure optimization of the inhibitor [78]. The structure of the binding site can also be used for pharmacophore modeling with further mining of the conformational databases [79]. Computational methods were used for the development of the inhibitors of SARS-CoV-2 PLpro [80,81]. Molecular docking of compounds from the ZINC drug database into the homology model of SARS-CoV-2 PLpro indicated a series of antivirus, antibacterial, muscle relaxant, and antitussive drugs that exhibit a high binding affinity to SARS-CoV-2 PLpro [82]. The nucleotide-like inhibitor of RdRp, ribavirin, was predicted to bind to the active site of the PLpro with the lowest binding energy. The predicted hydrogen bonds are between Gly164, Gln270, Tyr274, Asp303, and the compound. Additionally, π–π stacking was found between Tyr265 and the triazole ring in ribavirin [82]. Although ribavirin can cause anemia [51], it could be used as a starting point for the design of novel effective and safe CoV therapeutics. Several compounds from natural products have also been found to effectively bind with the protease [82]. Molecular docking into the homology model of SARS-CoV-2 PLpro allowed the identification of 16 FDA-approved drugs, including chloroquine and formoterol, which bind to the target enzyme with a significant affinity and good geometry [83]. Chloroquine is an effective drug approved for malaria treatment and has been investigated thoroughly [84]. Chloroquine was shown to interfere with the fusion and uncoating of HIV [85]. However, it was reported to inhibit autophagy and may potentiate tissue damage [86]. Another molecular docking of three inhibitors of SARS-CoV PLpro (GRL-0667, GRL-0617, and mycophenolic acid) and three inhibitors of HCV PLpro (telaprevir, boceprevir, and grazoprevir) revealed that each compound can bind to the active site of SARS-CoV-2 PLpro, which makes them prospective drugs [87,88]. Several compounds from Chinese medical herbs have also been found to bind with SARS-CoV-2 PLpro, 3CLpro, and S protein. The most effective potential inhibitors for CoV PLpro are cryptotanshinone, tanshinone IIa, and quercetin, which are effective against MERS-CoV and can be used in oral administration [89]. The docking of cyanobacterial metabolites into the active site of SARS-CoV-2 PLpro identified cryptophycin 1, cryptophycin 52, and deoxycylindrospermopsin as potential inhibitors. Subsequent molecular dynamics simulations and the assessment of the physicochemical properties and potential toxicity of the metabolites established deoxycylindrospermopsin as the most promising inhibitory candidate against both SARS-CoV-2 PLpro and 3CLpro [90]. The docking and molecular dynamics of the peptides of azurin, p18 and p28, identified their ability to bind SARS-CoV-2 PLpro. p28 also interacts with the human ACE-2 receptor, the S-protein, and 3CLpro [91]. Virtual screening of 1697 clinical FDA-approved drugs indicated several inhibitors active against SARS-CoV-2 PLpro. Phenformin, quercetin, and ritonavir were the most perspective [92]. The molecular docking and molecular dynamics of 97 antiviral secondary metabolites from fungi identified norquinadoline A and scedapin C as the most potent inhibitors for SARS-CoV-2 PLpro. Norquinadoline A conferred high gastrointestinal absorption, poor blood–brain barrier penetrability, and high drug-likeness as per Lipinski’s rule of five and did not demonstrate any toxicity [93].

Although computational methods permit the screening of millions of compounds in a relatively short period, several issues need consideration. These programs tend to stubbornly provide a result, no matter how absurd it is [94]. Docking is now routine in virtual screening for drug design. However, frequent problems arise, such as identification of the wrong binding site of the target protein, screening using an unsuitable small-molecule database, and lack of clarity over whether the compound is an inhibitor or agonist. Therefore, the researchers should provide a critical evaluation of their results [95]. Moreover, docking has a serious limitation, i.e., the structure of the target protein is usually rigid. This can be overcome at the cost of significant computing power. Therefore, the hit compounds obtained using virtual screening should be validated by the experiment.

The activity of previously identified naphthalene CoV inhibitors has assessed with regards to SARS-CoV-2 PLpro. In vitro using of fluorogenic peptide substrate indicated that the two most potent compounds were GRL0617 and compound 6, with IC_50_ values of 2.4 and 5.0 µM. The effectiveness was further proved in Vero E6 cells with EC_50_ values of 27.6 and 21.0 µM, respectively [96]. The effectiveness (IC50) of naphthalene inhibitor, GRL-0617, in another study reached 0.74 ± 0.07 µM against the DUB activity of SARS-CoV-2 PLpro and 0.66 ± 0.08 µM of SARS-CoV PLpro, although it did not inhibit MERS-CoV PLpro. The inhibitor fostered the antiviral interferon pathway and reduced viral replication in infected cells [97]. Other research designed fluorogenic peptides based on the substrate specificity of CoV PLpro. The fluorescent tag was further exchanged to a reactive group—vinyl methyl ester. The obtained inhibitors, VIR250 and VIR251, were active against CoV PLpro, but not against human DUBs [98]. Other novel inhibitors of CoV PLpro include the antioxidant, ebselen, and structural analogs. These compounds inhibited SARS-CoV-2 PLpro with an IC_50_ value of 0.236 µM [99]. A library of 5576 compounds comprising approved drugs and late-stage clinical drug candidates was checked using HTS for the ability to inhibit SARS-CoV-2 PLpro. Unfortunately, they were either ineffective or inhibited human deubiquitinase USP21 [100]. On the other hand, the noncovalent small molecule naphthalene SARS-CoV inhibitors, rac3j, rac3k, and rac5c, revealed low or sub-µM inhibitory activity against the DUB activity of SARS-CoV-2 PLpro. Rac5c also prevented self-processing of nsp3 (containing the PLpro domain) from the fusion protein nsp3-green fluorescent protein (GFP). Therefore, the inhibitor can potentially prevent the polyprotein processing. The compound decreased viral titer in Vero monkey kidney epithelial cells that approves its effectiveness [100]. *Strobilanthes cusia* extract and its components were checked for cytotoxicity and the inhibition of HCoV-NL63 activity and infectivity in LCC-MK2 cells. Tryptanthrin showed the strongest antiviral activity due to the inhibition of RdRp and PLpro and low cytotoxicity [101]. Moreover, 6-TG was recently shown to inhibit SARS-CoV-2 PLpro proteolytic and deISGylation activities in Vero-E6 cells and Calu3 cells at submicromolar levels [102].

A CoV PLpro inhibitor will not be effective unless it can be delivered to the cells infected by the virus. Since CoVs often induce respiratory tract infections, one might expect that inhibitors should be delivered to the lung. However, coronaviruses also affect the gastrointestinal tract, cardiovascular, central nervous systems, liver, and kidneys [103,104]. This is related to the fact that CoVs bind to the receptors that are ubiquitously expressed. However, intravenous injections possess certain risks, including anaphylactic reactions, infections, and embolism [105]. Moreover, the inhibitor has to get through the membrane to get into the infected cell. One prospective approach to deliver the drug is nanoparticles. These can enhance the stability, solubility, absorption, bioavailability, and controlled release of drugs [106]. So far, only siRNA-based therapeutics against CoV have been improved using nanoparticles [107]. The surface of the particle can be covered with antibodies to proteins on the cell surface or cell-penetrating peptides (CPPs). CPP can also be fused directly to the inhibitor to promote its penetration into a cell [108]. However, to the best of our knowledge, the delivery of the inhibitors for CoV PLpro has not been investigated yet although the outlined features impact pharmacokinetics and pharmacodynamics, which define the effectiveness and side effects of the drug [109].

## 7. Conclusions

CoVs are a biohazard, as we have seen in the cases of SARS-CoV, MERS-CoV, and SARS-CoV-2 outbreaks. Since the CoV genome tends to frequent mutations, more deadly CoVs may emerge in the future. One of the prospective approaches is to develop an inhibitor for an enzyme supporting the CoV life cycle. The folds of PLpro from different CoVs are similar, which enables the development of a broad-spectrum inhibitor. Moreover, PLpro has two important functions in the viral life cycle: it cleaves viral polyproteins and suppresses the host immune response. Thus, an inhibitor for PLpro will both prevent viral replication and boost the immune system of the host, contributing to the speedy recovery of the patient. Thus far, several inhibitors of CoV PLpro have been identified but none have entered clinical trials. This can be explained by the swiftness and small size of the previous outbreaks. The authors hope that the severity and rapid spread of SARS-CoV-2 will accelerate research into the existing inhibitors and identification of novel ones.

An inhibitor needs to suppress the proteolytic, deubiquitinating, and deISGylating activities of the PLpro simultaneously to ensure the suppression of viral replication and support the host immune response. Eventually, it will be necessary to develop drug delivery mechanisms that are suitable for different CoVs, which bind to distinct cell receptors. Virtual screening could enable these investigations to be conducted quickly. However, it is necessary to keep in mind its limitations and validate all results experimentally. Together, these approaches could lead to the development of broad-spectrum safe therapeutics, which simultaneously strike both the replication and immune response suppression of CoV.

## Figures and Tables

**Figure 1 pharmaceuticals-13-00277-f001:**
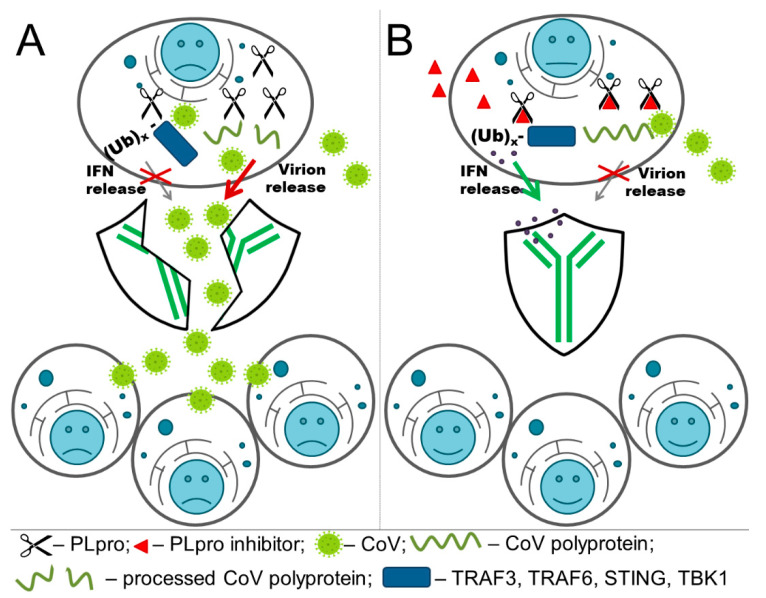
(**A**) Coronaviruses (CoV) papain-like proteases (PLpro) processes viral polyproteins, and thus promotes viral replication. Simultaneously, PLpro removes ubiquitin (Ub) chains and interferon-stimulated gene 15 product (ISG15) from the proteins involved in the antiviral immune response. Together, these factors produce viral spreading. (**B**) The inhibitor for CoV PLpro reduces the processing of the polyproteins, which prevent viral replication. Moreover, unimpaired immune response induces interferon (IFN) expression. All this results in suppression of viral spreading.

**Figure 2 pharmaceuticals-13-00277-f002:**
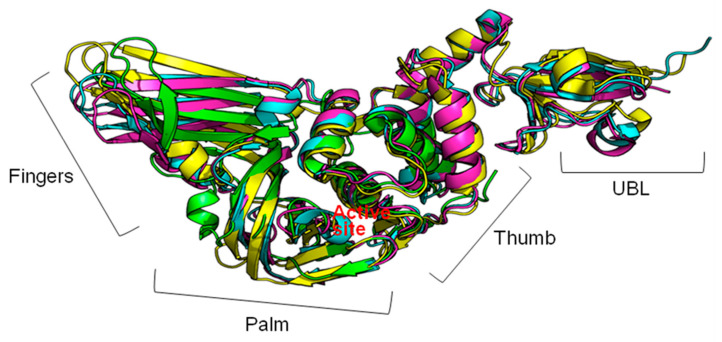
The figure shows the structures aligned in PyMOL of transmissible gastroenteritis virus (TGEV) PLpro (Protein Data Bank Identifier (PDB ID): 3MP2; green), Middle East respiratory syndrome CoV (MERS-CoV) PLpro (PDB: 4P16; yellow), severe acute respiratory syndrome coronavirus (SARS-CoV) PLpro (PDB: 2FE8; blue), and SARS-CoV-2 PLpro (PDB: 6W9C; pink); UBL—ubiquitin-like domain.

**Table 1 pharmaceuticals-13-00277-t001:** CoV and host enzymes supporting the viral life cycle.

Enzyme		Function
Viral enzymes		
Endoribonucleases ^1^	Nsp1	In complex with a 40S ribosome induces endonucleolytic cleavage near the 5′ untranslated region (UTR) of the host mRNA
	Nsp15	Possesses endoribonuclease (RNase) activity and enables the evasion of viral dsRNA from the host dsRNA sensors
Proteases	Nsp3 (up to two PLpro)	Process the viral polyproteins, deubiquitinate IRF3 and NF-κB, deISGylate IFN-stimulated gene 15 product (ISG15)
	Nsp5 (3CLpro)	Processes the viral polyproteins, signal transducer and activator of transcription 2 (STAT2), and NF-κB essential modulator (NEMO)
Primase	Nsp8	In complex with nsp7 synthesizes short RNA primers de novo
RNA-dependent RNA polymerase	Nsp12	In complex with nsp7 and nsp8 synthesizes RNA and enables replication and transcription of the viral genome
Helicase	Nsp13	Unwinds both double-stranded deoxyribonucleic acid (dsDNA) and dsRNA in a 5′-to-3′ direction and hydrolyzes deoxyribonucleotide triphosphates (dNTPs) and NTPs
Exoribonuclease, guanine-N7-methyltransferase	Nsp14	In complex with nsp10 possesses exoribonuclease and (guanine-N7)-methyltransferase (MTase) activities and methylates the RNA cap
2′-*O*-methyltransferase	Nsp16	In complex with nsp10 possesses 2′-*O*-MTase activity and methylates the RNA cap
Host enzymes		
Ribosomes		Together with factors involved in translation are recruited to translate viral genomic and subgenomic RNAs
Proteases	Serine (TMPRSS2, TMPRSS4, TMPRSS11a, TMPRSS13, HAT, trypsin, DESC1, elastase, factor Xa, plasmin, furin)	Cleave the S protein to promote the invasion of CoVs
	Cysteine (cathepsins B, L, and S)	

^1^ Anti-CoV inhibitors have been developed for the underlined enzymes.

**Table 2 pharmaceuticals-13-00277-t002:** Inhibitors for CoV PLpro.

Inhibitor	CoV	Inhibition of (IC_50_, µM)	Ref.
Pro ^1^	Ub ^2^	ISG15 ^3^
Naphthalene inhibitors 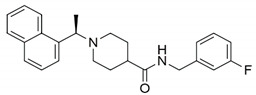	SARS-CoV-2	2.4 ± 0.02	0.74 ± 0.07	1.50 ± 0.08	[28,96,97,110]
SARS-CoV	0.15 ± 0.01	0.66 ± 0.08	0.66 ± 0.09
MERS-CoV	N.d. ^4^	N.d.	N.d.
Ebselen 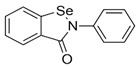	SARS-CoV-2	N.d.	2.02 ± 1.02	N.d.	[99]
SARS-CoV	N.d.	8.45 ± 0.96	N.d.
MERS-CoV	N.d.	N.d.	N.d..
Tanshinones from *S. miltiorrhiza* 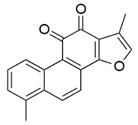	SARS-CoV-2	N.d.	N.d.	N.d.	[64]
SARS-CoV	0.8 ± 0.2	0.7 ± 0.2	N.d.
MERS-CoV	N.d.	N.d.	N.d.
Chalcones and coumarins from *A. keiskei* 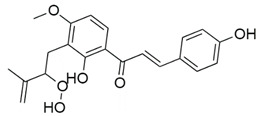	SARS-CoV-2	N.d.	N.d.	N.d.	[67]
SARS-CoV	1.2 ± 0.4	2.6 ± 0.7	1.1 ± 0.6
MERS-CoV	N.d.	N.d.	N.d.
Zn^2+^-ion and conjugates 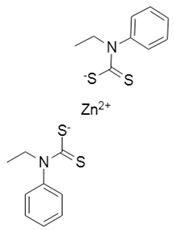	SARS-CoV-2	N.d.	N.d.	N.d.	[62]
SARS-CoV	1.3 ± 0.2	N.d.	N.d.
MERS-CoV	N.d.	N.d.	N.d.
Diarylheptanoids from *A. japonica* 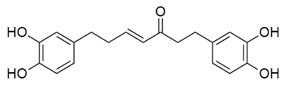	SARS-CoV-2	N.d.	N.d.	N.d.	[65]
SARS-CoV	4.1 ± 0.3	3.0 ± 1.1	N.d.
MERS-CoV	N.d.	N.d.	N.d.
Polyphenols from *B. papyrifera* 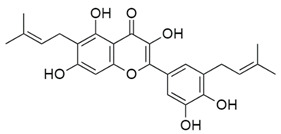	SARS-CoV-2	N.d.	N.d.	N.d.	[68]
SARS-CoV	3.7 ± 1.6	7.6 ± 0.4	8.5 ± 1.2
MERS-CoV	39.5 ± 5.1	N.d.	N.d.
N-Ethylmaleimide 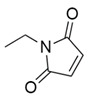	SARS-CoV-2	N.d.	N.d.	N.d.	[70,111]
SARS-CoV	N.d.	4.4 ± 1.0	N.d.
MERS-CoV	45.0 ± 10.1	N.d.	N.d.
Geranylated flavonoids from *P. tomentosa* 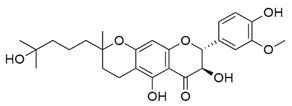	SARS-CoV-2	N.d.	N.d.	N.d.	[66]
SARS-CoV	5.0 ± 0.06	N.d.	N.d.
MERS-CoV	N.d.	N.d.	N.d.
Thiopurine compounds ^5^ 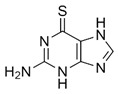	SARS-CoV-2	N.d.	N.d.	N.d.	[70,111]
SARS-CoV	N.d.	5.0 ± 1.7	N.d.
MERS-CoV	24.4 ± 4.3	12.4 ± 1.9	N.d.
8-(Trifluoromethyl)-9*H*-purin-6-amine 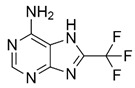	SARS-CoV-2	N.d.	N.d.	N.d.	[69]
SARS-CoV	10.9 ± 0.9	N.d.	N.d.
MERS-CoV	6.2 ± 0.9	N.d.	N.d.
Disulfiram 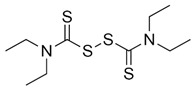	SARS-CoV-2	N.d.	N.d.	N.d.	[72]
SARS-CoV	14.2 ± 0.5	24.1 ± 1.8	N.d.
MERS-CoV	22.7 ± 0.5	14.6 ± 0.3	N.d.

^1^ Proteolytic activity; ^2^ deubiquitinating activity; ^3^ deISGylating activity; ^4^ not determined; ^5^ FDA-approved drugs indicated in green.

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
