# Peer review of "Papain-Like Proteases as Coronaviral Drug Targets: Current Inhibitors, Opportunities, and Limitations"

_pharmaceuticals, 2020, doi:10.3390/ph13100277_

Round 1

Reviewer 1 Report

This is a comprehensive review summarizing not only the functional roles, structural features, and the inhibitors for coronaviral Papain-like protease (PLpro), but also the biological functions of other viral proteins and the inhibitors for other viral enzymes. It is a timely review during this COVID-19 outbreak.

Two suggestions for the review:

  1. The abstract can be modified to explicitly state what is covered in this review, such as the last sentence in the abstract can be changed to: “This review summarizes the structural features of CoV PLpro, the inhibitors that have been identified over the last twenty years, and the compounds that have the potential to become novel effective therapeutics against CoVs in the near future.”
  2. Figure 1 is important in describing the essential functions of PLpro in viral spreading. However, this figure is not clear and difficult to understand.

Author Response

Dear Editor and Reviewers,

Thank you for your very helpful suggestions. We have fully addressed your concerns below and havethoroughly revised the manuscript based on your recommendations. All changes have been marked in yellow.

This is a comprehensive review summarizing not only the functional roles, structural features, and the inhibitors for coronaviral Papain-like protease (PLpro), but also the biological functions of other viral proteins and the inhibitors for other viral enzymes. It is a timely review during this COVID-19 outbreak.

Two suggestions for the review:

  1. The abstract can be modified to explicitly state what is covered in this review, such as the last sentence in the abstract can be changed to: “This review summarizes the structural features of CoVPLpro, the inhibitors that have been identified over the last twenty years, and the compounds that have the potential to become novel effective therapeutics against CoVs in the near future.”

Author response: We thank the reviewer for this observation. The sentence has been corrected (lines 20-22).

  1. Figure 1 is important in describing the essential functions of PLpro in viral spreading. However, this figure is not clear and difficult to understand.

Author response: We thank the reviewer for this observation. We did our best to make Figure 1 more comprehensible (line 174).

Reviewer 2 Report

The review article is complehansThe review article has a comprehensive literature survey on the coronavirus PLpro. The article could add the data about the inhibitory effect of CoV PLpro inhibitors on the replication of coronavirus in vitro, in vivo and in clinic. 

Author Response

Dear Editor and Reviewers,

Thank you for your very helpful suggestions. We have fully addressed your concerns below and havethoroughly revised the manuscript based on your recommendations. All changes have been marked in yellow.

The review article is complehansThe review article has a comprehensive literature survey on the coronavirus PLpro. The article could add the data about the inhibitory effect of CoVPLpro inhibitors on the replication of coronavirus in vitro, in vivo and in clinic. 

Author response: We thank the reviewer for her/his high assessment of the manuscript.